# Fluorescence Polarization Immunoassay for Determination of Enrofloxacin in Pork Liver and Chicken

**DOI:** 10.3390/molecules24244462

**Published:** 2019-12-05

**Authors:** Xing Shen, Jiahong Chen, Shuwei Lv, Xiulan Sun, Boris B. Dzantiev, Sergei A. Eremin, Anatoly V. Zherdev, Jianfa Xu, Yuanming Sun, Hongtao Lei

**Affiliations:** 1Guangdong Provincial Key Laboratory of Food Quality and Safety, South China Agricultural University, Guangzhou 510642, China; shenxing325@163.com (X.S.); chenjiahong2009@163.com (J.C.); ciweihw@163.com (S.L.); lingdubingxing@126.com (J.X.); gzsyming@163.com (Y.S.); 2State Key Laboratory of Food Science and Technology, School of Food Science of Jiangnan University, Wuxi 214122, China; sxlzzz@jiangnan.edu.cn; 3A.N. Bach Institute of Biochemistry, Research Center of Biotechnology of the Russian Academy of Sciences, Leninsky prospect 33, 119071 Moscow, Russia; dzantiev@inbi.ras.ru (B.B.D.); saeremin@gmail.com (S.A.E.); zherdev@inbi.ras.ru (A.V.Z.); 4Chemical Department, M. V. Lomonosov Moscow State University, Leninskie Gory, 119991 Moscow, Russia

**Keywords:** enrofloxacin, fluorescence polarization immunoassay, antibody, pork liver, chicken

## Abstract

Enrofloxacin (ENR) is a widely used fluoroquinolone (FQ) antibiotic for antibacterial treatment of edible animal. In this study, a rapid and highly specific fluorescence polarization immunoassay (FPIA) was developed for monitoring ENR residues in animal foods. First, ENR was covalently coupled to bovine serum albumin (BSA) to produce specific polyclonal antibodies (pAbs). Three fluorescein-labeled ENR tracers (A, B, and C) with different spacers were synthesized and compared to obtain higher sensitivity. Tracer C with the longest arm showed the best sensitivity among the three tracers. The developed FPIA method showed an IC_50_ (50% inhibitory concentration) of 21.49 ng·mL^−1^ with a dynamic working range (IC_20_–IC_80_) of 4.30–107.46 ng·mL^−1^ and a limit of detection (LOD, IC_10_) of 1.68 ng·mL^−1^. The cross-reactivity (CR) of several structurally related compounds was less than 2%. The recoveries of spiked pork liver and chicken samples varied from 91.3% to 112.9%, and the average coefficients of variation were less than 3.83% and 5.13%, respectively. The immunoassay took only 8 min excluding sample pretreatment. This indicated that the established method had high sensitivity, specificity, and the advantages of simplicity. Therefore, the proposed FPIA provided a useful screening method for the rapid detection of ENR residues in pork liver and chicken.

## 1. Introduction

Fluoroquinolone (FQ) drugs are widely used in the practices of veterinary clinics due to their effective and broad-spectrum antibacterial activity [1]. Enrofloxacin (ENR) is an important member of this family, and exhibits an excellent activity against mycoplasma as well as Gram-positive and Gram-negative pathogens by affecting DNA gyrase [2]. As a third-generation fluoroquinolone only available in veterinary medicine, enrofloxacin is thus used in many species with few adverse effects. Enrofloxacin has also been used as growth promoters in livestock and poultry production for the purpose of pursuing maximum economic benefits, resulting in a dramatic increase in its consumption [3]. In parallel to the exposure to low levels of these antibiotics, an increase has been observed in human pathogen resistance constituting a public health hazard, primarily through the increased risk of treatment failures [4,5]. In addition, enrofloxacin residues can adversely affect the human digestive system and nervous system as well as cause allergic reactions [6]. Therefore, the European Union and the Ministry of Agriculture of the People’s Republic of China have set the maximum residue limits (MRLs) of enrofloxacin in muscle tissue to 100 μg**·**kg^−1^, and the MRLs in the kidneys to 200 μg·kg^−1^ (No. 278, 22 May 2003), while Japan has set MRL in chicken muscle at 10 μg·kg^−1^ [7,8]. However, monitoring enrofloxacin residues in treated animal foods remains a challenge.

Although several instrumental methods have been reported for monitoring enrofloxacin residues, including high-performance liquid chromatography (HPLC) and surface plasmon resonance (SPR) biosensor technologies [4,9,10,11,12], they are unable to meet the increasingly growing demand for rapid screening since these methods are time-consuming, low-throughput, laborious, and in need of expensive instruments [13].

For the rapid screening of enrofloxacin, immunochemical methods, especially enzyme-linked immunosorbent assay (ELISA) [7,8,14] and chemiluminescent enzyme immunoassay (CLEIA) [15,16,17], have been successfully applied to official and laboratory purposes on account of their high sensitivity. However, as a heterogeneous assay, ELISA has a multistep and time-consuming process. To overcome the above problems, the fluorescence polarization immunoassay (FPIA)-based method, which simplifies the current immunoassay for routine applications, transforms the method from heterogeneous (separation step) to homogeneous (no separation step). The FPIA meets the requirements of a simple, reliable, fast, and cost-effective analytical method, shortening the determination of the analyte within a very short period of time [18]. Some studies have been reported for the detection of FQs by FPIA [19,20,21]. In our previous study, heterologous tracers were recruited to establish FPIA for the detection of six FQs based on anti-clinfloxacin polyclonal antibody (pAb) [22]. However, since enrofloxacin is more popular than other FQs [15], specific immunoassays are needed to accurately quantify enrofloxacin residues.

In this study, a FPIA method for the rapid analysis of enrofloxacin in animal food samples was developed. Three fluorescent tracers with different spacers were synthesized and compared. The extraction conditions and sample matrix effect on FPIA performance were investigated and optimized.

## 2. Results and Discussion

### 2.1. Immunoreagent Preparation

The development of an immunoassay with good analytical properties mainly depends on the design of appropriate hapten and the selection of a conjugation site to the carrier, in order to induce available antibodies [23,24]. For the preparation of artificial antigen, the carbonyl group of enrofloxacin was coupled to bovine serum albumin (BSA) and the resulting enrofloxacin-BSA immunogen was confirmed by using UV spectroscopy (Figure 1). The absorbance peaks of the enrofloxacin appeared at 271 nm and 315 nm, whereas the peak of BSA was at 280 nm. As expected, the absorbance peaks of artificial antigen were different from those of enrofloxacin and BSA, shifted toward the middle of the two materials. This indicated that the coupling of enrofloxacin to BSA was successful [7]. Two rabbits were injected with enrofloxacin-BSA immunogen, and both produced the final antisera of high titer values after the fifth injection (fourth boost). Therefore, the antisera of two rabbits were mixed for the development of FPIA.

In this study, enrofloxacin was covalently attached to three different fluoresceins (AF, EDF, and HDF, see details in Materials and Methods) through chloroacetamide moiety, leading to a different length of the conjugation [23,25]. The resulted tracers at R*_f_* = 0.2, R*_f_* = 0.7, and R*_f_* = 0.5 were confirmed by HPLC-MS (ESI-MS (positive), *m*/*z* 689.2, 791.2, and 847.3 [M + H]^+^, see Appendix A), respectively, which indicated that tracers A (ENR-AF), B (ENR-EDF), and C (ENR-HDF) were successfully synthesized. The binding abilities of the three tracers to the antibody were subsequently verified as shown in Figure 2. The results indicated all three tracers showed sufficient binding capability with the employed antibody against enrofloxacin.

### 2.2. Optimization of FPIA

It is known that suitable working concentrations of antibody and competitive antigen are essential to enhance the sensitivity of a competitive immunoassay [26,27]. In order to establish a highly sensitive and specific FPIA method, the antibody concentration for FPIA is usually selected to be an unsaturated level that results in about 70% binding rate to ensure a strong enough signal and good sensitivity [7,28]. In the present study, different concentrations (0.1, 0.5, 1.0, and 2.0 nmol·L^−1^) of three tracers and the dilution time of antibody were evaluated by three parameters (IC_50_, δmP, and δmP/IC_50_) to improve the FPIA sensitivity. All three tracers exhibited the highest δmP/ IC_50_ value at 0.5 nmol·L^−1^. Results at the optimal concentration are shown in Table 1. Titer represents the amount of a specific antibody present in the serum. δmP means the difference between the maximal and minimal fluorescence polarization signals. IC_50_ means the analyte concentration that produces 50% inhibition of tracer binding in the FP assay. The limit of detection (LOD, IC_10_) and the dynamic working range (IC_20_–IC_80_) were defined as the analyte concentration that inhibited 10% and 20–80% of tracer binding, respectively. It showed that 0.5 nmol·L^−1^ tracer C exhibited the highest δmP (124), lowest IC_50_ (21.8 ng·mL^−1^), and maximal δmP/IC_50_ (5.68). Therefore, tracer C with a concentration of 0.5 nmol·L^−1^ was selected as the optimal tracer.

The competitive kinetic curves of tracer C are shown in Figure 3. The kinetic reaction tends to be stable after 8 min. This shows that signals were significantly different under different concentrations even after only 1 min, from when it could be considered as recordable for the homogeneous immunoassay. However, a longer time was desired for manual operation to minimize the inter-assay variation coefficient, due to successively changing of the signal at the earlier stage. Thus, 8 min was selected in the present FPIA.

Figure 4 shows a typical optimal calibration curve. After 8 min incubation, the LOD (IC_10_), IC_50_, and dynamic working range (IC_20_–IC_50_) for enrofloxacin detection with 0.5 nmol·L^−1^ tracer C were 1.68 ng·mL^−1^, 21.49 ng·mL^−1^, and 4.30–107.46 ng·mL^−1^, respectively. The performance of this FPIA is good enough to meet the detection requirements for enrofloxacin [5,7]. Although some reported immunoassays showed an extreme sensitivity, like CLEIA [15] and fluorescent nanoparticle-based immunochromatographic assay (FN-ICA) [29], the practical uses of the methods were limited by their narrow linear range (Table 2). In comparison, the detection range for other immunoassays including indirect competitive ELISA (icELISA) [7], competitive fluorescence-linked immunosorbent assay (cFLISA) [30], and this FPIA is suitable for enrofloxacin MRL level. Moreover, the established FPIA has simpler operational steps resulting in a shorter time.

### 2.3. Specificity

To study the specificity of obtained antibody, the cross-reactivity (CR) was tested against a group of compounds structurally related to enrofloxacin. The results are shown in Table 3. The CR to tested FQs was less than 2%, which indicated that the developed FPIA had high specificity to enrofloxacin.

The only structural difference between enrofloxacin and ciprofloxacin is the substituent at the 4′-position of the piperazine ring, an ethyl for enrofloxacin while there is a H-atom for ciprofloxacin, whereas the activities are significantly different from each other. The substitute groups at the 4′-position of the other four compounds are a methyl for ofloxacin (also for levofloxacin), a H-atom together with a methyl for gatifloxacin, and no piperazine ring in flumequine. In spite of the structural differences on the main ring, all these related compounds have very low CR values. Therefore, it can be assumed that the ethyl in piperazine, as the distal group of hapten from the coupling site, may be recognized by the antibody as a characteristic epitope. When the substitute group in ring piperazine was changed from ethyl to methyl, or H-atom, the cross-reactivity decreased more than 56-fold from 100% to 1.8%, or 83 folds from 100% to 1.2%, respectively. Moreover, for gatifloxacin, one more methyl group was added at the 5-position of the piperazine ring and the CR dropped to 0.6%. The flumequine, which lacks a piperazine, has no cross-reactivity at all. All of these indicated that the 4′-position is of great importance in the antibody recognition.

### 2.4. Matrix Effect

As a rapid screening tool, the pretreatment in FPIA of animal tissue samples consisted of a simply extraction without a cleanup step. It can be predicted that the pH value of animal tissue extracts may change the fluorescence-label structure [31], resulting in instability of the mP value. The ionic strength existing in the animal tissue fluid, as well as the organic solvent brought in by the extraction step, may affect the binding affinity between the antibody and analyte, leading to a decrease in assay sensitivity [23,32]. Thus, the matrix effects of animal tissue samples on the enrofloxacin FPIA sensitivity were investigated through the effects of pH, NaCl, and water-miscible organic solvent methanol.

pH effect. In comparison with acidic conditions, the calibration curves obtained under alkali and neutral conditions showed satisfactory analytical parameters with lower IC_50_ and better δmP/IC_50_ value (Figure 5A1,A2). As the fluorescent tracer was synthesized from fluorescein isothiocyanate (FITC), similar results were obtained by Ma’s research [31]. Under different pH, the FITC molecule shows diverse structures, which produce different fluorescence quantum yields. The relative fluorescence intensity of FITC increased with a rise in pH under 10.0 in Ma’s research. However, in the present study, the combining capacity decreased under higher alkalic conditions due to their impact on antibody stability [32]. Therefore, pH 8.0 borate buffer (BB) with the lowest IC_50_ and the best δmP/IC_50_ was selected for further investigation.

Tolerance of NaCl. One of the potential interfering factors of FPIA is the inorganic ions in the animal tissue sample. However, as the main ions in animal tissue, the NaCl concentration has no significant adverse effect on the measured performance of the BB solution even up to 3.4% (*v*/*v*). The sensitivity (IC_50_ and δmP/IC_50_) and signal intensity (δmP) were not decreased until the NaCl concentration was higher than 10% (Figure 5B1,B2). Although high concentrations of NaCl can severely reduce the combined ability due to the antibody denaturation, the NaCl density in animal tissues (0.85% physiological saline) is much lower than the tolerance range of FPIA (3.4%) under similar conditions. Therefore, animal tissue samples can be directly assayed by developed FPIA without further dilution.

Tolerance of organic solvent. As shown in Figure 5C1,C2, the IC_50_ of the assay increased as the methanol concentration increased, and δmP decreased significantly in the absence of methanol. Therefore, even with only 5% (*v*/*v*) in BB, methanol significantly decreased the sensitivity of FPIA. This is consistent with the fact that, as described in previous studies [23,24], higher concentrations of methanol could denature the antibody, thereby inhibiting its binding to the tracer, resulting in the generation of a lower signal. Therefore, the introduction of methanol into the samples should be avoided, and another organic solvent (ethyl acetate) was used as an extraction solvent for enrofloxacin extraction in animal tissue samples.

Real sample matrix effect. Real animal tissue matrices may affect assay sensitivity. Their effects were evaluated by comparing the calibration curves of actual samples (pork liver and chicken) and BB. The result showed that the calibration curves were superposed to a large extent (data not shown), implying that the simple sample pretreatment method could basically eliminate the matrix interference.

### 2.5. Recovery

Under the conditions described for the developed FPIA method (see 3.4 FPIA Procedure), the mean recovery of enrofloxacin from the pork liver and chicken samples was 105.7% and 95.6%, respectively (Table 4), and the mean coefficients of variation (CVs) for pork liver and chicken were 3.83% and 5.13%, respectively. This indicated that the recovery and reproducibility of the developed FPIA were satisfactory for a rapid screening analysis.

## 3. Materials and Methods

### 3.1. Reagents and Instruments

Chemical reagents and organic solvents were of analytical grade unless otherwise specified. Bovine serum albumin (BSA), 1-ethyl-3-(3-dimethyl-aminopropyl) carbodiimide hydrochloride (EDC), *N*-hydroxysuccinimide (NHS), *N*,*N*-dime-thylformamide (DMF), isobutyl chloroformate, tributylamine, fluorescein isothiocyanate isomer I (FITC), 5-aminofluorescein (AF), and complete and incomplete Freund’s adjuvants were all purchased from Sigma-Aldrich (St. Louis, MO, USA). Ethylenediamine (EDA) and hexamethylenediamine (HDA) were obtained from Weijia Company (Guangzhou, China). Silica gel G glass sheets (type GF254, layer thickness 0.2 mm) for thin layer chromatography (TLC) were purchased from Taizhou Shenghua Material Corporation (Zhejiang, China). Microplates (96 wells) were obtained from Jinchanhua Corporation (Shenzhen, China). The enrofloxacin and the cross-reactants (norfloxacin, ciprofloxacin hydrochloride, ofloxacin, levofloxacin, gatifloxacin, and flumequine) of analytical standard were gifts received from the Veterinary College of South China Agricultural University (Guangdong, China). Pork liver and chicken samples were purchased from a local supermarket (Guangdong, China).

Borate buffer (BB, 50 mmol, pH 8.5) with 0.01% sodium azide was used as working buffer for all FPIA experiments. The enrofloxacin stock solutions (20 mg/mL) and other related cross-reactants were prepared by dissolving 20 mg of each in 1 mL of methanol and were stored at −20 °C before use. Standard solutions of analytes in the range of 0.10–50,000 ng·mL^−1^ were prepared by dilution of stock solution with BB. Standard solutions were stored in methanol at 4 °C before use.

FPIA was conducted on a multilabel counter (Wallac 1420 Victor^3^, PerkinElmer Company, Waltham, MA, USA). Centrifugation operation was carried out on a Sigma 6K 15 centrifuge (Beijing BMI Instruments Co. Ltd., Beijing, China). Ultraviolet (UV) spectral data was recorded on a UV spectrophotometer (Hitachi, Kyoto, Japan).

### 3.2. Generation of Polyclonal Antibody

Enrofloxacin was coupled to BSA by carbodiimide-coupling method and characterized by UV, as previously described [33]. Briefly, 16 mg (0.24 μmol) BSA and 10 mg (27 μmol) enrofloxacin were mixed in 2 mL phosphate buffer (PBS, pH 7.4, 0.01 mol·L^−1^), followed by the addition of 24 mg (150 μmol) EDC, then mixed thoroughly and kept at room temperature overnight. The mixture was dialyzed against PBS (pH 7.4, 0.01 mol·L^−1^) for three days with three changes per day. The dialyzed product was centrifuged at 12,000 rpm for 10 min. The supernatant was then collected as immunogen and stored at −20 °C until use. The artificial antigen, BSA, and enrofloxacin were compared by UV-Vis spectra.

Two New Zealand rabbits weighing 1.5–2.0 kg were immunized 4 times using enrofloxacin-BSA as the immunogen at intervals of three weeks. After the first immunization, rabbit blood was collected eight days after each injection, and tested for the detection of enrofloxacin antibodies by indirect ELISA [34]. The polyclonal antibody obtained was divided into aliquots (1 mL), labeled, and stored at −20 °C until use. The pre-immune serum was obtained from the rabbit before the first immunization as negative control.

### 3.3. Preparation of Fluorescein-Labeled Enrofloxacin Tracers

Two fluorometric reagents, fluorescein thiocarbamyl ethylenediamine (EDF) and fluorescein thiocarbamyl hexylenediamine (HDF), were prepared using the similar procedures as described by Wang [35]. Briefly, enrofloxacin (3.6 mg, 10 μmol) and isobutyl chlorocarbonate (1.6 μL, 12 μmol) were dissolved in DMF (300 μL) containing triethylamine (3 μL), and the mixture was stirred for 20 min in an ice bath. Then, the fluorometric reagent, AF (5 mg, 10 μmol), EDF (4.5 mg, 10 μmol), or HDF (5 mg, 10 μmol) was added to the supernatant to synthesize Tracer A, B, or C, respectively (Figure 6). After the color of the solution had changed to yellow, a small portion of the reaction mixture (approximately 50 μL) was subjected to TLC using chloroform, methanol, and formylic acid (4:0.4:0.04, *v*/*v*/*v*) as the eluent. The concentration of the tracer was determined spectrophotometrically at a wavelength of 492 nm, and it was assumed that the absorbance in BB buffer (0.2 mol/L, pH 8.0) was the same for fluorescein (ε = 8.78 × 104 L/(mol·cm)) [36]. The chemical structures of the obtained tracers were confirmed by HPLC-MS.

### 3.4. FPIA Procedure

FPIA calibration curves were obtained by adding 100 μL of the tracer solution in BB and 20 μL calibrator solution or sample to the microplate well, and then 100 μL of the optimal dilution of antiserum were added to the mixed well. The fluorescence polarization (FP) of the reaction mixture was analyzed and plotted against the concentration of analyte. The fluorescence polarization signal (mP) was measured and plotted against the logarithm of analyte concentration, following the same procedure described by Lei [23]. The four-parameter logistic equations in OriginPro7.0 software package (OriginLab, Northampton, MA, USA) as defined below were used to fit the immunoassay data:y= (A − D)/[1 + (X/C)^B^] + D(1)
where A represents at high asymptote, D is response at low asymptote, B is slope factor, C is concentration corresponding to 50% specific binding, and X is calibration concentration. The IC_50_ value was defined as the analyte concentration that produces 50% inhibition of tracer binding in the FP assay. The limit of detection (LOD, IC_10_) and the dynamic working range (IC_20_–IC_80_) were defined as the analyte concentration that inhibited 10% and 20–80% of tracer binding, respectively [35].

### 3.5. Optimization of Immunoreagents

Immunoreagent conditions, including the concentration of antibody and tracer, were optimized to improve the sensitivity of FPIA. Briefly, to construct the antibody dilution curves, the three tracers were diluted (0.1, 0.5, 1.0, and 2.0 nmol·L^−1^) and then mixed with the serially diluted antibody (from 1/100 to 1/12,800) in BB. The antibody dilution titer, which showed 70% binding response to the tracer, was chosen as the working concentration [7,28]. Subsequently, to optimize the tracer concentration, tracers were diluted and mixed with respective antibody at working concentration to construct a series of calibration curves to evaluate δmP, IC_50_, and δmP/IC_50_, where δmP is the difference between the maximal and minimal fluorescence polarization signals. Lower IC_50_ values and higher values of δmP and δmP/IC_50_ indicate a more sensitive assay.

### 3.6. Kinetics of FPIA

To achieve the optimal incubation time, a competitive kinetic curve was plotted. Amounts of 100 µL of tracer C solution (0.5 nmol·L^−1^) and 20 µL of standard solution (0, 10, 100, 1000 ng·mL^−1^, respectively) were mixed in the microplate well. Then, 100 µL of antibody (dilution of 1:600) were added into the mix well and the mP value versus time was recorded successively from 1 min.

### 3.7. Specificity

Six FQs were selected for the specificity investigation of the FPIA under optimized FPIA conditions. According to the following equation, cross-reactivity (CR) was calculated, where IC_50_ is the concentration at which 50% of the enrofloxacin antibody is bound to the analyte:CR% = [IC_50_ (enrofloxacin)/IC_50_ (structurally related compounds)] × 100%(2)

### 3.8. Recovery Matrix Effects

Several physicochemical factors influencing the FPIA were studied. Modifications of δmP and IC_50_ parameters of the calibration curves were evaluated under different conditions. Buffer pH and ionic concentration were optimized using tracers with higher sensitivity. For buffer pH, competitive curves were performed with buffers of different pH values in constant ionic concentration of 200 mmol. A stock solution of 200 mmol boracic acid (pH = 5.4) and 200 mmol borate sodium (pH = 9.5) were prepared. The two stock solutions were mixed to reach desired buffer pH. For buffer ionic and organic solvent effects, calibration curves were performed with buffers of different ionic and organic solvent concentration values in constant pH 8.0.

The factors, including pH and ionic strengths of dissolved salt and organic solvent, were studied to assess the potential matrix interference of real samples. The pH value of BB was tested for compatibility with the ELISA from 6.6 to 9.0. Inorganic salt (NaCl in BB) was tested from 0 to 20% (*v*/*v*). Organic solvent (methanol) was tested from 0 to 50% (*v*/*v*). Matrix effects were evaluated using enrofloxacin-free samples (pork liver and chicken) purchased from a local supermarket.

### 3.9. Sample Extraction and Recovery Study

To evaluate the recovery of the developed FPIA, pork liver and chicken were both spiked with enrofloxacin and the recoveries were determined by FPIA. Negative samples were confirmed at the Guangzhou Institute of Food Inspection, and the positive samples were spiked with enrofloxacin on the basis of negative ones. The pork liver samples were cut into small pieces and homogenized. Enrofloxacin standard was added to the 1 g samples to the final concentrations of 5, 10 and 50 μg·kg^−1^, and the mixture was stirred for 2 min. Then, a 10 mL amount of ethyl acetate was added to samples and shaken for 30 min. After being centrifuged at 4500*g* for 10 min, the supernatant was transferred into another centrifuge tube. The sample was re-extracted two more times using the same method, and the final resulting supernatant was evaporated with a nitrogen gas drier at 50 °C until oily residue remained. The residue was re-dissolved by adding 2 mL PBS buffer and vortex-mixed for 2 min, and then the extraction solution was diluted 1:3 in BB before being used in the FPIA. Blank samples were prepared as described above but were not spiked with analyte. Finally, the enrofloxacin concentration in spiked samples was detected by the established FPIA.

## 4. Conclusions

A rapid and highly specific FPIA was developed for the first time to detect enrofloxacin in pork liver and chicken. The IC_50_ value of the FPIA was 21.49 ng·mL^−1^, and the LOD was 1.68 ng·mL^−1^. Compared to other methods (e.g., conventional HPLC (LOD is 7.99 ng·mL^−1^) and icELISA (LOD is 0.2 ng·mL^−1^) mentioned above), the established method demonstrated comprehensive advantage as higher sensitivity than HPLC and simpler operation than ELISA. Furthermore, this FPIA exhibited high specificity to enrofloxacin and a very short loading time. Therefore, the proposed FPIA provided a useful screening method for the rapid detection of enrofloxacin residues in pork liver and chicken muscles.

## Figures and Tables

**Figure 1 molecules-24-04462-f001:**
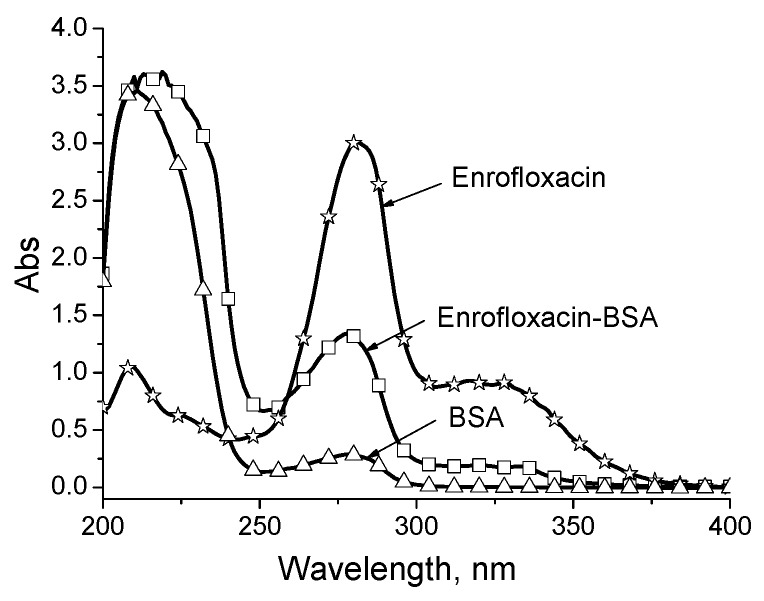
Spectra of enrofloxacin, bovine serum albumin (BSA), and enrofloxacin-BSA.

**Figure 2 molecules-24-04462-f002:**
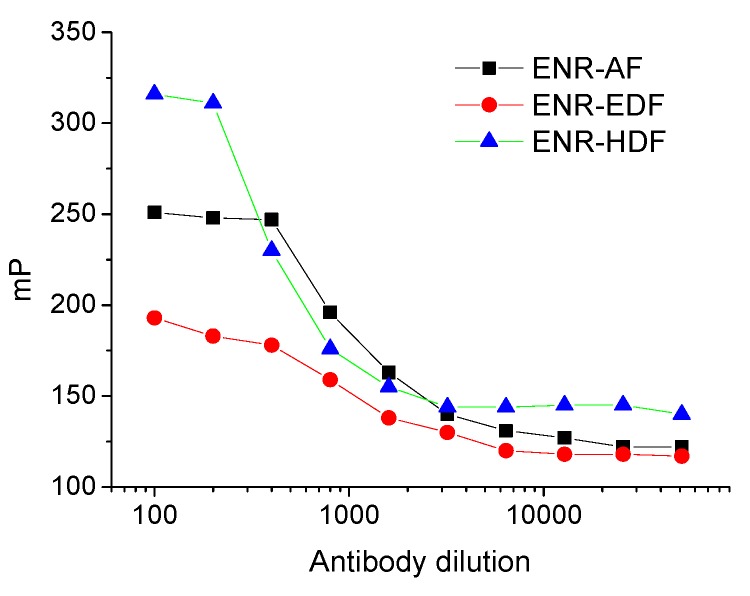
Antibody dilution curve against three fluorescent tracers.

**Figure 3 molecules-24-04462-f003:**
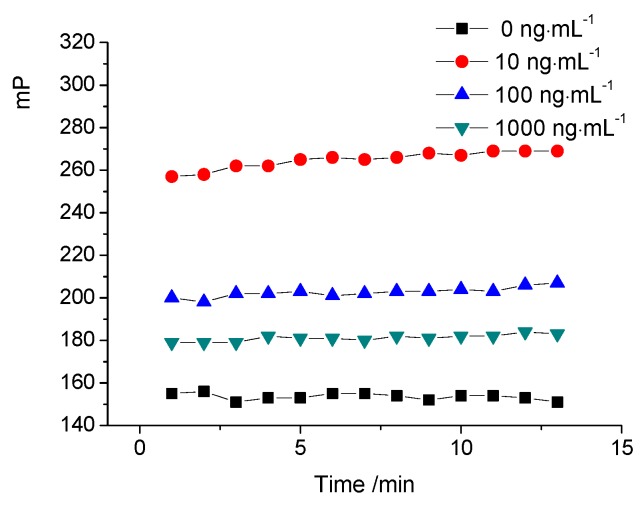
Competitive kinetic curves of tracer C and antibody.

**Figure 4 molecules-24-04462-f004:**
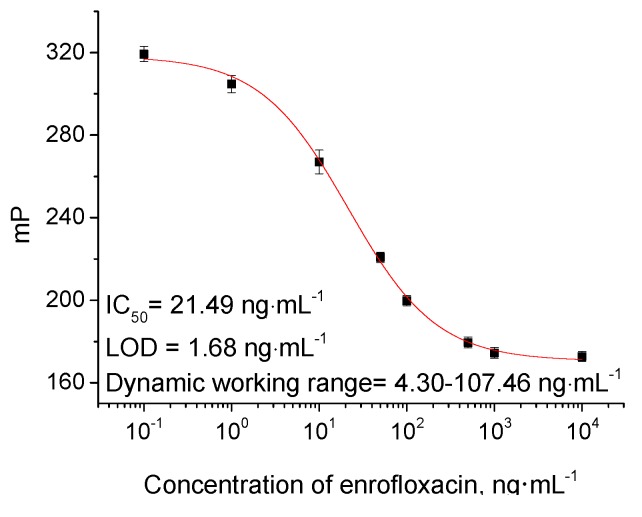
FPIA calibration curve for enrofloxacin with tracer C. Each point represents mean ± standard deviation of three replicates.

**Figure 5 molecules-24-04462-f005:**
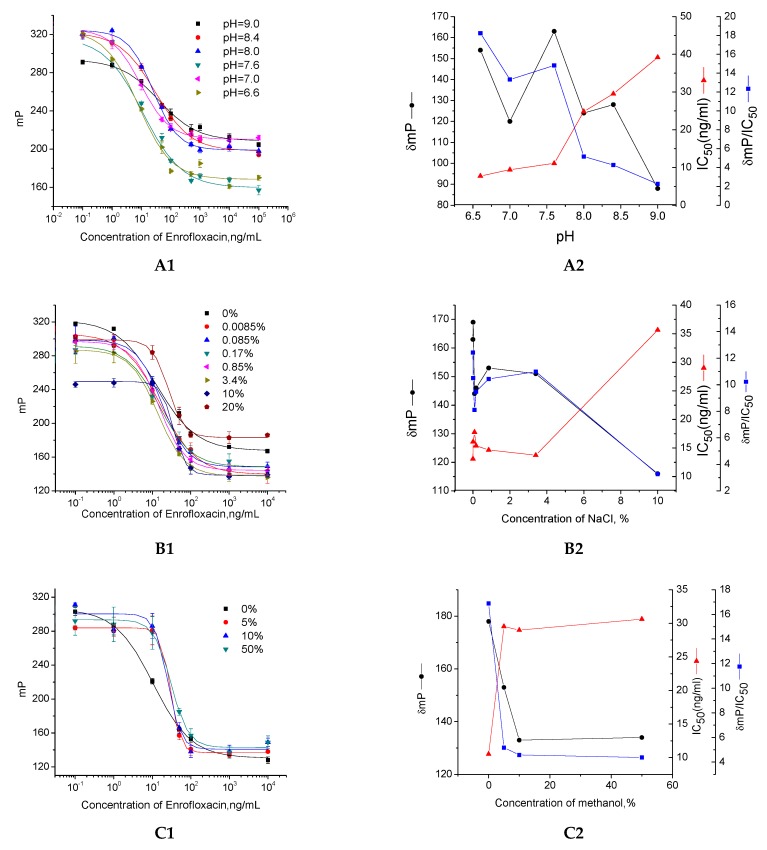
Effect of pH, ionic salt, and organic solvent on FPIA. Each point represents mean ± standard deviation of three replicates. (**A1**,**A2**) Effect of pH from 6.6 to 9.0 on FPIA. (**B1**,**B2**) Effect of ionic salt (NaCl) from 0 to 20% (*v*/*v*) on FPIA. (**C1**,**C2**) Effect of organic solvent (methanol) from 0 to 50% (*v*/*v*) on FPIA.

**Figure 6 molecules-24-04462-f006:**
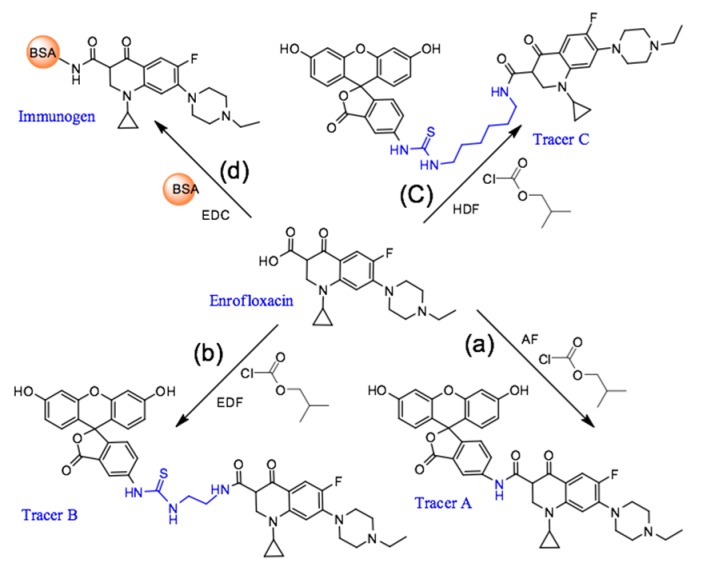
Synthesis of the fluorescein-labeled tracer and immunogen. (**a**) Tracer A; (**b**) tracer B; (**c**) tracer C; (**d**) immunogen.

**Table 1 molecules-24-04462-t001:** Analytical characteristics of fluorescence polarization immunoassay (FPIA) with various tracers (at the optimal concentration of 0.5 nmol·L^−1^).

Tracer	Titer	δmP	IC_50_ (ng·mL^−1^)	LOD (ng·mL^−1^)	Dynamic Working Range (ng·mL^−1^)	δmP/IC_50_
Tracer A	1/600	89	49.93	2.01	6.59–378.48	1.74
Tracer B	1/300	39	9.34	0.75	1.80–48.60	4.18
Tracer C	1/300	124	21.82	3.85	5.04–108.33	5.68

**Table 2 molecules-24-04462-t002:** Comparison among different detection methods of enrofloxacin.

Method	LOD (ng·mL^−1^)	LOQ (ng·mL^−1^)	IC_50_ (ng·mL^−1^)	Linear Range (ng·mL^−1^)	Testing Samples	Reference
HPLC	7.99	26.6	-	-	Milk	Huang, 2018 [12]
SPR	0.3	-	3.21	-	Milk	Fernández, 2010 [10]
icELISA	0.2	0.6	9.4	0.6–148.0	Beef, pork	Zhang, 2011 [7]
CLEIA	0.03	0.35	-	0.35–1.0	Milk, eggs, honey	Yu, 2014 [15]
cFLISA	2.5	-	8.3	1–100	Chicken	Chen, 2009 [30]
FN-ICA	0.02	-	0.22	0.025–3.5	Chicken	Huang, 2013 [29]

**Table 3 molecules-24-04462-t003:** Cross-reactivity of polyclonal antibody (pAb)-enrofloxacin to related compounds based on tracer C.

No.	Compound	Structure	IC_50_(nmol·mL^−1^)	CR (%)
Moiety	Substituent Group
**1**	Enrofloxacin	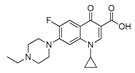	ethyl	0.027	100.0
**2**	Ofloxacin	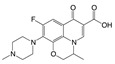	methyl	1.50	1.8
**3**	Levofloxacin	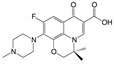	methyl	1.62	1.6
**4**	Ciprofloxacin	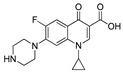	H	2.21	1.2
**5**	Gatifloxacin	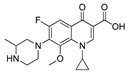	H	4.65	0.6
**6**	Flumequine	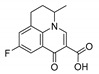	H	ND ^a^	≤0.01

^a^ ND (not determined), presented infinite IC_50_ values and could not be fitted with the four-parameter logistic equation.

**Table 4 molecules-24-04462-t004:** Recovery of enrofloxacin from spiked pork liver and chicken samples (*n* = 3).

Sample	Spiked Level (μg·kg^−1^)	Observed Value (μg·kg^−1^)	Recovery (%, *n* = 3)	Mean Recovery (%)	CV (%)	Mean CV (%)
Pork liver	5	5.64 ± 0.46	112.9 ± 9.1	105.7	8.07	3.83
	10	10.46 ± 0.33	104.6 ± 3.3		3.13	
	50	49.79 ± 0.15	99.6 ± 0.3		0.30	
Chicken	5	4.56 ± 0.31	91.3 ± 6.2	95.6	6.76	5.13
	10	10.36 ± 0.26	103.6 ± 2.6		2.47	
	50	45.99 ± 2.84	92.0 ± 5.7		6.17	

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
