# Peer review of "Fluorescence Polarization Immunoassay for Determination of Enrofloxacin in Pork Liver and Chicken"

_molecules, 2019, doi:10.3390/molecules24244462_

Round 1

Reviewer 1 Report

In the manuscript “Fluorescence polarization immunoassay for determination of enrofloxacin in pork liver and chicken” the authors present a fluorescent assay based on the FPIA approach to detect in real matrices this antibiotic compound.

In my opinion, this work is of interest to the readers of the Journal but I recommend the publication after major revision. In particular:

The manuscript lacks the following experiments:

1. In the manuscript, the authors declare that they develop a polyclonal antibody against enrofloxacin staring from the enrofloxacin-BSA conjugate antigen. Following this activity the manuscript lacks the following experiments:
a) Indirect ELISA shows the binding capability of the produced antibody.
b) Indirect ELISA tests performed on similar molecules (Ciprofloxacin, ofloxacin, etc.) to demonstrate the cross-reactivity between the different compounds.

So, to allow the reader to understand the results the authors should include and comment on these data (such as a table, etc.) in the manuscript. Please, the authors provide it in the new version of the manuscript.

2. The authors have produced labeled enrofloxacin using a different fluorescent probe. For this section the following information lacks and should be described in the new version of the manuscript:

a). The authors mentioned that chemical structures of the labeled enrofloxacin were been confirmed by HPLC-MS but few details in the manuscript are reported. Please, the authors include this characterization in the new version of the manuscript (such as in supplementary materials).

b). Have the conjugation of the enrofloxacin to the fluorescent probe (AF, EDF, and HDF) an effect on the optical properties of the molecules? A spectrum of the fluorescent probes and the fluorescent molecules (enrofloxacin-AF, enrofloxacin-EDF, and enrofloxacin-HDF) produced should be included in the manuscript for example as supplementary materials. Which is the effect of the conjugation on the binding capability of the produced antibody?

3. In the manuscript, the authors have shown experiments in real matrices (pork and chicken) and describe the percentage of the recovery. Which is the positive and negative control used in these real matrices experiments? Please, the authors provide it.

4. More details are needed in the legend of figure 3 and figure 4. Please, the authors provide it.

Author Response

Reply to reviewer 1

Thanks for the reviewer’s comments. We have carefully revised this manuscript following the reviewer’s suggestions as below:

1: The reviewer said indirect ELISA assays should be included in the manuscript in order to investigate the activity of the polyclonal antibody against enrofloxacin. According to the review’s comment, we add dilution curves of the antibody with different tracers as new Fig. 2 to make an additional explanation of antibody activity. However, in common situation, ELISA assays are not adopted in the construction of FPIA method, like other reports (Kolosova et al, Talanta 2017, 162: 495-504) (Beloglazova and Eremin, Talanta 2015, 142: 170-175) (Borduleva et al, Anal Bioanal Chem 2018, 410: 6923-6934). Besides, the FPIA calibration curve for enrofloxacin and the cross-reactivities to different compounds have already demonstrated the activity of the antibody.

2: About the labeled fluorescent tracers, the reviewer has two comments:

The reviewer said that more details about structure confirmation of the labeled enrofloxacin should be provided. For this suggestion, we provide the mass spectrogram for the synthesized tracers as supplemental materials. The reviewer wondered if the conjugation of the enrofloxacin to the fluorescent probe has effect on the optical properties of the tracers and the binding capability of the antibody. Of course, the conjugation may have effect on optical properties of the synthesized molecules. However, the selection of the best fluorescent tracer for developing FPIA method is eventually relied on the binding capability to the antibody, and these results have already been shown in Table 1 in the form of IC50 and LOD values for different tracers. The optical properties of fluorescent molecules themselves are usually not tested in FPIA method development (Kolosova et al, Talanta 2017, 162: 495-504) (Zvereva et al, J Pharmaceut Biomed 2018, 159: 326-330)). To calculate the concentration of the fluorescent molecules, UV spectrum was employed as a common method (Yakovleva et al, Anal Bioanal Chem 2004, 378: 634-641). As the reviewer suggested, we add “Negative samples were confirmed by Guangzhou Institute of Food Inspection, and the positive samples were spiked with enrofloxacin on the basis of negative ones” at line 304-306 in “3.9 sample extraction and recovery study” section, to state the source of the negative and positive samples in real matrice expriments. Follow the reviewer’s comment, now more information is added in the legends of Fig. 4 (named as Fig. 3 before revised) and Fig. 5 (named as Fig. 4 before revised).

Reviewer 2 Report

Shen et al has described the development of “Fluorescence polarization immunoassay for determination of enrofloxacin in pork liver and chicken”. However, there are lots of points to be considered before the acceptance of the manuscript:

Provide a comparison table for the other methods of same analyte to their LOD, LOQ values. The paper requires a thorough English correction. The language and use of punctuation marks are not correct like at page 1 in introduction (line 41)- it must be either interacting or affecting DNA gyrase not the effecting DNA gyrase. It is better to explain the mechanism with proper references. The units of concentration must be changed to ng mL-1 from ng/mL throughout manuscript. Introduction- line 49. No detail of which Ministry of Agriculture author taking about. No need of repetition of Maximum residue limits again in line 50 when abbreviation has already given in line 49. Similarly, author can use ENR abbreviation further. Please cross check the SPR reference for high throughput comparison. Line 70- it must be animal food samples. Line 87- Provide full meaning of AF, EDF and HDF when used first time in manuscript. Line 89- “which indicated that the tracer A………” Line 90- it must be “successfully synthesized”. Line 116-119- it is not clear; re-writing is required. Line 119- which regulation author is referencing…explain. I think better to explain sigmoidal fit model instead of linear fit- as I can see from Fig 3 it is sigmoidal fit as per one side saturation model. Line 124- it is table 2 or table 3, please check. Line 128- I think numbering of heterocyclic ring is wrong- it must be 4’. Line 154- similar result obtained be Ma et al. compare the results in one or two lines. What is BB in line 156. It must be selected not chosen in line 156. Typo graphical error 165-170. Line 173-177- this text does not reflect in Fig. 4 and figure legend is incomplete. Line 179- resulting in the generation of a lower signal. Table 3- what is µp/Kg and µb/Kg- no similarity. Explain? Line 198- Chemical reagents…… Line 208- analytical gifts received from the Vet…….. Line 223- full meaning of PBS as it is written first time. Line 229- was collected not taken. Line 244- it must be BB buffer not borate buffer. 3.4 check language – avoid repetition. Rephrasing required. Line 266- diluted antibody in what medie? Section 3.7- start with Six FQs not 6 kinds of ….. Line 290- How it can be diluted to 400 mmol when the stock is 200 mmol only? Line 296- how the author sure that the sample were ENR free? Are they certified sample from regulatory agency? Conclusion no comparison provided with existing literature. Reference pattern need to be same either abbreviation or full form. Ref 6- only one-page number is provided. 15 what P.d.B- kindly check. Paper need thorough typographical error for space, comma, dots in the manuscript.

Author Response

Reply to reviewer 2

Thanks for the comments provided by the reviewer. We have revised this manuscript throughout following the reviewer’s suggestions.

A comparison table for the other methods of enrofloxacin is added in the introduction. Line 41, “…effecting DNA gyrase” is revised to “…affecting DNA gyrase”, and the bacteriostatic mechanism of enrofloxacin is supported by reference 1. All the units of ng/mL and nmol/L are changed into ngmL-1 and nmol·L-1 throughout the manuscript. Line 49, the name of “the Ministry of Agriculture of the People’s Republic of China” is complemented. The repeated full name of “Maximum residue limits” is deleted in line 51. However, the name of “enrofloxacin” is kept instead of its abbreviation ENR to be consistent with other compounds. High throughput is not suitable to describe the method developed in this work due to enrofloxacin is the only research object. We have already adjusted “high-throughput screening” to “rapid screening” at line 61 in the introduction. In this situation, SPR with multichannel is not included here. Line 74, “food samples” is revised to “animal food samples”. The full meaning of AF, EDF and HDF were introduced in the sections of “1 Reagents and instrcments” and “3.3 Preparation of Fluorescein- Labeled Enrofloxacin Tracers” in Material and Methods. Line 97 “which indicated that the tracer A…” and line 98 “successfully synthesized” are revised following the reviewer’s comments. Line 131-134 has been re-written. We have explained the sigmoidal fit model but not linear fit model for text description. The linear fit chart was also provided for the reason of considering the reading habits of some readers. “Table 2” is changed to “Table 3” now at line 141. We corrected all the numbers of the figures and tables. Line 145, the number of the substitute position was corrected to 4’ from 4-position. Similar pH effect was obtained by Ma et al. We have added the comparison of the results. BB solution mentioned in line 175 is a common abbreviation for borate buffer, which is now added when it first appears, as the reviewer suggested. Also, the word “chose” is changed to “selected” in line 176. For the explanation of Fig. 5 (used to be Fig.4), line 198-203 has been re-written. The units for both spiked level and observed level are μg·Kg-1, and it has been corrected in Table 4 (used to be table 3). The antibody was diluted with BB buffer, which has been added at line 290. For the matrix effect study, line 311-313 has been re-written. The ENR free samples were confirmed by Guangzhou Institute of Food Inspection, we add this description at line 321-323. Comparison between established FPIA and other methods is complemented to the conclusion section as the reviewer suggested. Reference pattern is re-organized.

Besides, we have made a through English correction and other minor revisions as the reviewer suggested:

Line 40, “pocess an excellent activity” is revised to “exhibits an excellent activity”.

Line 219, use “Chemical reagents” instead of “General reagents”.

Line 229, “…are gifts from…” is corrected to “…are gifts received from…”.

Line 244, the full name of PBS is added.

Line 229, “rabbit blood was taken…” is changed into “rabbit blood was collected…”.

Line 251, “borate buffer” is changed into “BB buffer”.

Line 272, “3.4 Fluorescence polarization immunoassay procedure” is shortened into “FPIA procedure”.

Line 302, “6 kinds of…” is changed into “Six FQs…”.

Reviewer 3 Report

Fluorescence polarization immunoassay for determination of enrofloxacin in pork liver and chicken

Manuscript ID: molecules-624662

Shen et al. report the development of a fluorescence polarization immunoassay (FPIA) for the detection of enrofloxacin (ENR) in pork liver and chicken spiked with ENR. This study shows a very high resemblance to the authors’ previous work (J. Sci. Food Agric. 2016, 96, p. 1341) and does not provide any novelty in terms of method development. Similar studies for the detection of fluoroquinolones by FPIA were also reported by other authors, however, they are not cited in the manuscript (e.g. 10.1039/C3AY42034E). (A good review that discusses antibiotic detection in raw meat with different analytical methods should be cited as well: 10.3390/antibiotics6040034).

 Except for the novelty issue, there are some points that need to be addressed in order to improve the clarity of the manuscript:

Page 2, Line 81: The authors mention ‘OVA’ at the end of the sentence. I believe this is ovalbumin, however, nothing has been mentioned regarding the ovalbumin in the paper. Page 3, Lines 87-88: there are several abbreviations that are not explained. Please write in full when they first appear in the text. AF, EDF, HDF, Rf, etc. Page 3: Please move the paragraph (Lines 103-107) at line 98, after the sentence that ends as ‘…FPIA sensitivity.’ Page 4, ‘Specificity’: was the cross-reactivity against structurally-related compounds tested in the presence or absence of ENR? It would be interesting to test specificity in a mixture of all related compounds. Page 5: What is BB? Only at the experimental section, the BB is specified, however, it comes after the Results & Discussion section. Please write in full when they first appear in the text. Page 6: Figure 4 B1 and C1: Figure legends are not clear. What do % values refer to? Please specify as ‘xxx % NaCl’ or organic solvent, etc. Page 7, Line 190: In the first sentence of the ‘Recovery’ section, please specify the ‘conditions described above’. A complementary technique such as HPLC should be used for validation of the results. The manuscript requires language editing and grammar correction.

Author Response

Reply to reviewer 3

The reviewer said this study shows a very high resemblance to our previous work. Thanks for the reviewer paying attention to our previous study. Although this work and the previous work (J Sci. Food Agric.) both developed a fluorescence polarization immunoassay (FPIA) for fluoroquinolones, there are several novelty for method development in this work. Firstly, as is known that the tracer structure could greatly influence the immunoassay performance. In the previous work, the amino group of 6 kinds of fluoroquinolones were exposed and used to synthesize 6 kinds of fluorescein-labeled tracers with fluorescein isothiocyanate isomer I, respectively. In this work, the carboxyl group of enrofloxacin was exposed and synthesized three fluorescent tracers with different spacers, and the effects of different fluorescent tracer structure on the performance of the FPIA were investigated, which was not studied in previous work (J Sci. Food Agric.). Secondly, a competitive kinetic curve was plotted to achieve the optimal incubation time, which was not showed in the previous work (J Sci. Food Agric.). At last, while the liquid sample (goat milk) was spiked with clinafloxacin in the previous work, the recovery study for enrofloxacin in the solid sample (pork liver and chicken) were investigated in this study. And the effects of extraction conditions and several physicochemical factors (pH, ionic strengths of dissolved salt and organic solvent) which influencing the FPIA were investigated and optimized. Besides, some studies for the detection of fluoroquinolones are cited (reference 1, 19-21), following the reviewer’s suggestion. About the “OVA” mentioned in the “2.1 Immunoreagent preparation”, we are sorry for making this mistake. Line 83-84 has been re-written without “OVA”. The full meaning of AF, EDF and HDF were introduced in the sections of “1 Reagents and instrcments” and “3.3 Preparation of Fluorescein- Labeled Enrofloxacin Tracers” in Material and Methods. To avoid repeat, we now add a “see details in Method and Materials” after the abbreviations to guide readers. The paragraph (line 113-117) has already moved into the paragraph above, as the reviewer suggested. About the specificity discussion, we didn’t test the cross-reactivity against structurally related compounds in the presence of ENR. We tested the activity individually for each compound, and calculated the relative ratio to that of enrofloxacin, following the classic calculation method of cross-reactivity. So far as we know, there’s no appropriate method to evaluate the mixture. BB is the abbreviation of “borate buffer”, which is added when it first appears in the line 175, as the reviewer suggested. Follow the reviewer’s comment, now more information is added in the legend of and Fig. 5 (named as Fig. 4 before revised)。 The method mentioned in “Recovery” section was described in “4 FPIA procedure”. We add the explanation at line 210. HPLC and SPR methods used for validation of enrofloxacin detection had been carried out at the same time. However, these materials are organized in other manuscript under submission. We have made an English correction throughout as the reviewer suggested. We hope it will be satisfactory.

Round 2

Reviewer 1 Report

Authors have addressed all the major issues in the revised version of manuscript. The manuscript has been improved and it can be published in its current form.

Author Response

Thanks for the comments provided by the reviewer.

Reviewer 2 Report

Table 1- Comparison table can be improved further and add present work comparison too in the table. Add space between Tabel 1 and the comparison word. Line 84- space between 271nm Line 95- Rf must be Rf I still think it's better to explain the Sigmoidal fit model compare to the linear fit. Because sigmoidal fit looking good to me and according to one-side binding assay it may be the present case. The author can check it once. Line 193- it must be under similar conditions. Table 4- space in the value provided with a standard deviation Line 234- It can be 0.10- 50,000 ng mL-1. Throughout the manuscript, the author can check once and change the value to significant figures that were not provided e.g. 0.10 ng mL-1. In the case like 7.99 ng mL-1 is fine. Check the sub and superscript in the manuscript. Once check the MRLs provided by other regulatory bodies also.

Author Response

Thanks a lot for the careful revision of this manuscript, and we appreciate for the comments provided by the review. The reply for the comments are listed below:

Table 1- Comparison table can be improved further and add present work comparison too in the table.

Reply: We have improved table 1 with more references. In consideration of the position of table 1 (in the introduction), we didn’t add the results of the present work in it. However, we added the comparison between the established FPIA and other methods in 2.3 section at line 136 to 142.

It's better to explain the Sigmoidal fit model compare to the linear fit. Because sigmoidal fit looking good and according to one-side binding assay it may be the present case. The author can check it once.

Reply: We are sorry that we didn't make it clear for the explanation of the calibration curve before. Actually, the results showed here including the values of LOD, IC50 and linear range, were all obtained from the Sigmoidal fit model, not linear model. The linear fit chart was only provided for the convenience of some readers who are more used to the linear model. Now we have deleted the linear fit chart to avoid misunderstanding.

Throughout the manuscript, the author can check once and change the value to significant figures that were not provided e.g. 0.10 ng mL-1. In the case like 7.99 ng mL-1 is fine.

Reply: Follow the reviewer’s comment, we have checked and modified the significant figures for the values throughout the manuscript.

Once check the MRLs provided by other regulatory bodies also.

Reply: We have added the MRL set by Japan at line 51, and thus the references were re-organized.

The other format errors like space missing and sub and superscript in the manuscript have been checked and revised according to the reviewer’s comments.

Reviewer 3 Report

Authors have addressed all the major issues in the revised version of manuscript. The claritiy and quality of the manuscript has been improved and it can be published in its current form.

Author Response

(The authors gave the same response as above.)
